# Brain Pericytes Enhance MFSD2A Expression and Plasma Membrane Localization in Brain Endothelial Cells Through the PDGF-BB/PDGFRβ Signaling Pathway

**DOI:** 10.3390/ijms26135949

**Published:** 2025-06-20

**Authors:** Takuro Iwao, Fuyuko Takata, Hisataka Aridome, Miho Yasunaga, Miki Yokoya, Junko Mizoguchi, Shinya Dohgu

**Affiliations:** Department of Pharmaceutical Care and Health Sciences, Faculty of Pharmaceutical Sciences, Fukuoka University, 8-19-1 Nanakuma, Jonan-ku, Fukuoka 814-0180, Japan; t.iwao.ot@adm.fukuoka-u.ac.jp (T.I.); ftakata@fukuoka-u.ac.jp (F.T.); pd191001@cis.fukuoka-u.ac.jp (H.A.); pd211010@cis.fukuoka-u.ac.jp (M.Y.); pd211011@cis.fukuoka-u.ac.jp (M.Y.); pd231004@cis.fukuoka-u.ac.jp (J.M.)

**Keywords:** blood–brain barrier (BBB), pericytes, brain endothelial cells, MFSD2A, PDGF-BB/PDGFRβ signaling, docosahexaenoic acid

## Abstract

The brain actively obtains nutrients through various transporters on brain microvessel endothelial cells (BMECs). Major facilitator superfamily domain–containing protein 2a (MFSD2A) serves as a key transporter of docosahexaenoic acid (DHA) at the blood–brain barrier (BBB) and is exclusively expressed in BMECs. Although brain pericytes (PCs) regulate MFSD2A expression in BMECs, the underlying mechanism remains unclear. To determine whether PDGF-BB/PDGFRβ signaling between endothelial cells (ECs) and PCs affects MFSD2A protein expression and plasma membrane localization in ECs, we examined the impact of AG1296 (a PDGF receptor inhibitor) and *Pdgfrb*-knockdown PCs on a non-contact coculture BBB model comprising the primary cultures of rat brain ECs and PCs. The effects of PCs on MFSD2A expression, localization, and brain endothelial DHA uptake was assessed using Western blot, immunofluorescence staining, and [^14^C]DHA uptake by ECs, respectively. In ECs cocultured with PCs, MFSD2A expression and plasma membrane localization were significantly higher than in EC monolayers. Moreover, conditioned medium derived from PCs failed to enhance MFSD2A expression. The increased expression and membrane localization of MFSD2A were inhibited by AG1296 and *Pdgfrb*-knockdown PCs. Furthermore, PCs significantly increased [^14^C]DHA uptake by ECs. These findings suggest that PCs enhance MFSD2A expression and plasma membrane localization in ECs through PDGF-BB/PDGFRβ signaling.

## 1. Introduction

The central nervous system (CNS) is protected by the blood–brain barrier (BBB), which is composed of brain microvessel endothelial cells (BMECs), astrocytes, and pericytes (PCs). The BBB plays a crucial role in maintaining brain homeostasis by restricting the passage of substances from the bloodstream into the brain. Tight junction-associated proteins in BMECs are essential for regulating paracellular permeability. Moreover, the BBB restricts paracellular permeability and regulates transcellular permeability, including the transport of ions, neurotransmitters, and nutrients from blood to the brain through various transporters expressed on BMECs [1,2]. In various neurological diseases, including chronic Alzheimer’s disease and Parkinson’s disease, and acute ischemic stroke, a traumatic brain injury, epileptic seizures, and brain tumors, the BBB undergoes functional and structural breakdown [3,4].

Major facilitator superfamily domain-containing protein-2a (MFSD2A), a member of the major facilitator superfamily of membrane proteins, is exclusively expressed in BMECs [5,6]. It serves as the primary transporter mediating the uptake of lysophosphatidylcholine-docosahexaenoic acid (DHA) into the brain [7] and also facilitates the uptake of non-esterified (NE)-DHA by brain endothelial cells (ECs) [8]. The proper localization of MFSD2A in BMECs is essential for transport [6,7]. Beyond its role in DHA transport in the brain, MFSD2A contributes to BBB formation and function [6,9]. In particular, MFSD2A helps in maintaining a low-level permeability of the BBB to regulate transcytosis in BMECs [10]. MFSD2A is also associated with the pathology of diseases impacting the nervous system by regulating BBB integrity [11,12].

PCs are present at intervals along capillary walls and surround BMECs in the CNS. They interact with neighboring cells, including BMECs and astrocytes, and contribute to CNS homeostasis by regulating cerebral blood flow, stabilizing BBB integrity, and mediating immune responses [13,14]. Increased transcytosis was observed in cerebral vascular endothelial cells isolated from Mfsd2a knockout mice, in which pericytes appeared normal [6]. In contrast, pericyte-deficient mice exhibited a significant decrease in MFSD2A expression, indicating that pericytes are necessary for MFSD2A expression [6]. However, the mechanism through which PCs regulate MFSD2A expression in BMECs remains unclear.

In this study, we aimed to investigate how PCs regulate MFSD2A expression and localization in BMECs.

## 2. Results

### 2.1. PCs Regulate MFSD2A Expression on Brain ECs via Cell–Cell Interaction

To evaluate the effect of PCs on MFSD2A expression in ECs, MFSD2A protein expression levels in ECs were assessed using non-contact coculture BBB models comprising both ECs and PCs. Figure 1a presents a schematic of the experimental procedure using these non-contact coculture models. First, PCs were seeded at different densities in 24-well transwell plates and cultured with or without ECs for 3 d in vitro (DIV). MFSD2A protein expression levels in the PC coculture group were 35.1% (*p* = 0.0113) and 28.2% (*p* = 0.0437) higher at PC:EC ratios of 1:10 and 1:5, respectively, compared to those in the EC monolayer group. One-way ANOVA showed a significant effect of coculturing with PCs on MFSD2A protein expression (Figure 1b; F = 4.996, *p* = 0.0143). Next, we examined whether the PC-derived soluble factors affect MFSD2A protein expression in ECs. Figure 1c presents a schematic of the experimental procedure used to evaluate the effects of soluble factors released by intact PCs on ECs. The conditioned medium derived from PCs or cell-free cultures was collected, and ECs were cultured for 3 d in each medium. The PC-conditioned medium did not alter the MFSD2A protein expression levels in ECs. PDGFRβ is exclusively expressed in perivascular PCs and commonly used as a characteristic marker for PCs [15,16]. A previous study showed that MFSD2A expression dramatically decreased in Pdgfb^ret/ret^ mice, which exhibits a major loss of pericyte coverage [6]. Therefore, we hypothesized that PDGFRβ, as a receptor for PDGF-BB released from ECs, plays a critical role in regulating MFSD2A expression. To determine whether PDGF-BB/PDGFRβ signaling between ECs and PCs regulates MFSD2A protein expression in ECs, we evaluated the effects of the tyrosine kinase inhibitor AG1296 on MFSD2A protein levels in ECs. AG1296 binds to the intracellular domain of the PDGF receptor. Figure 1e presents a schematic of the experimental procedure used for evaluating the effect of AG1296 on MFSD2A protein levels in ECs. MFSD2A protein expression in the vehicle-treated PC coculture group was 54.2% (*p* = 0.0027) higher than that in the vehicle-treated EC monolayer group, and this increase was inhibited (*p* = 0.0363) following AG1296 treatment (Figure 1f). These findings indicate that EC–PC interaction enhances MFSD2A protein expression in ECs.

### 2.2. PDGF-BB/PDGFRβ Signaling Between Brain ECs and PCs Regulates MFSD2A Expression on Brain ECs

Although AG1296 is a potent and selective inhibitor of PDGFR kinase, it may also inhibit other kinases, including PDGFRα, Bek (FGF receptor) tyrosine kinase, and c-kit, in addition to PDGFRβ. Therefore, we further investigated whether PDGFRβ in pericytes contributes to regulating MFSD2A expression in ECs. To further validate the involvement of PDGF-BB/PDGFRβ signaling between ECs and PCs in MFSD2A protein expression, we examined the effect of siPdgfrb transfection into PCs on MFSD2A protein expression in ECs. Figure 2a shows the experimental procedure for siPdgfrb transfection into PCs. First, we evaluated the impact of the transfection duration on siPdgfrb transfection efficiency. Transfection with siPdgfrb significantly affected the PDGFRβ protein expression levels in PCs. However, the siPdgfrb transfection efficiency was independent of the transfection period (Figure 2b). Two-way ANOVA revealed significant effects of siPdgfrb transfection (F (1, 12) = 135.4, *p* = 0.0100) but no significant impact of the transfection period (F (2, 12) = 0.5881, *p* = 0.3863) and the interaction between siPdgfrb transfection and the transfection period (F (2, 12) = 1.835, *p* = 0.2017). The protein expression levels of PDGFRβ in the siPdgfrb groups on different days following transfection were 82.26 (1 DIV, *p* = 0.0006), 124.1 (2 DIV, *p* < 0.0001), and 105.5% (3 DIV, *p* < 0.0001) higher than those in the control group. Next, to validate the duration of Pdgfrb knockdown in PCs, cells were transfected with siPdgfrb for 1 DIV and the medium was replaced with EC medium, followed by culturing for 3 DIV. The PDGFRβ protein expression levels were 56% lower (*p* = 0.0001) in the siPdgfrb groups than in the control group (Figure 2c). This result indicated that the effect of Pdgfrb knockdown in PCs was sustained for at least 3 DIV. We then evaluated the impact of *Pdgfrb*-knockdown PCs on MFSD2A protein expression in ECs. Figure 2d shows the experimental procedure used to assess this effect. MFSD2A protein expression in the PC (control) coculture group was significantly higher than that in the EC monolayer group by 26.6% (*p* = 0.0050), and this increase was inhibited (*p* = 0.0004) by *Pdgfrb* knockdown in PCs (Figure 2e).

### 2.3. PCs Increase Plasma Membrane Localization of MFSD2A in Brain ECs

A previous study using immuno-electron microscopy confirmed that MFSD2A is localized on the plasma membrane in cerebral cortex capillaries [6]. Therefore, next we investigated whether PCs regulate MFSD2A protein expression levels and their localization to the plasma membrane in ECs. As illustrated in the schematic diagram (Figure 3a), ECs were cultured with or without PCs for 1 or 3 DIV, followed by immunofluorescence staining for MFSD2A and occludin. The plasma membrane of ECs was identified by the immunoreactivity of occludin, a tight junction-associated protein localized at the plasma membrane. MFSD2A localized to the plasma membrane was assessed by quantifying the MFSD2A and occludin double-positive regions. In ECs cocultured with PC for 3 DIV, the MFSD2A and occludin double-positive area was 2.1-fold (*p* = 0.0076) higher than in the EC monolayer at 3 DIV (Figure 3b,c). Two-way ANOVA revealed the significant effects of coculturing with PCs (F(1, 11) = 9.978, *p* = 0.0091), but no significant effects of time (F(1, 11) = 3.250, *p* = 0.0989) or the interaction between the coculture condition and time period (F(1, 11) = 3.250, *p* = 0.0989).

### 2.4. PDGF-BB/PDGFRβ Signaling Between Brain ECs and PCs Regulates MFSD2A Plasma Membrane Localization in Brain ECs

Next, to determine whether PDGF-BB/PDGFRβ signaling between brain ECs and PCs mediates the plasma membrane localization of MFSD2A in ECs, non-contact cocultured BBB models were treated with AG1296, as shown in Figure 4a. The MFSD2A and occludin double-positive region in the vehicle-treated PC coculture group was 212.4% higher than that in the vehicle-treated EC monolayer group (*p* = 0.0040). AG1296 significantly inhibited the PC coculture-induced increase in the MFSD2A and occludin double-positive area (*p* = 0.0213 vs. the vehicle-treated PC coculture); however, AG1296 showed no significant effect in the EC monolayer group (Figure 4b,c).

### 2.5. PCs Upregulate DHA Uptake by Brain ECs

We examined whether PCs enhance DHA uptake by brain ECs. We previously demonstrated that MFSD2A mediated non-esterified DHA uptake by brain ECs [8]. Figure 5a illustrates the experimental procedure used to evaluate the cellular uptake of [^14^C]DHA as non-esterified DHA by ECs in a non-contact cocultured BBB model. The uptake of [^14^C]DHA by ECs in the PC coculture group (the ratio of PC/EC, 1/2.5) was 30% (*p* = 0.0191) higher than that in the EC monolayer group (Figure 5b). One-way ANOVA revealed a significant effect of the PC coculture on [^14^C]DHA uptake by ECs (F = 3.946, *p* = 0.0191). Despite a significant increase in MFSD2A expression at a PC:EC ratio of 1:5, as shown in Figure 1b, no significant increase in [^14^C]DHA uptake by ECs was observed. The significantly increased uptake was only detectable at a PC:EC ratio of 1:2.5.

## 3. Discussion

Pericytes (PCs) play a critical role in the expression of MFSD2A in brain microvessel endothelial cells (BMECs) [6]; however, the mechanism underlying this regulation remains unclear. Given its increasing recognition as a key regulator of BBB integrity in the pathology of neurological diseases [11,12], elucidating the mechanism regulating MFSD2A expression is crucial. In this study, we investigated how PCs regulate the expression and localization of MFSD2A in BMECs.

Using a non-contact coculture BBB model comprising ECs and PCs, we demonstrated that ECs cocultured with PCs significantly increased MFSD2A protein expression compared to ECs in a monoculture (Figure 1). Although increased MFSD2A expression in ECs was not dependent on the number of PCs (Figure 1), our results suggest that the presence of soluble factors released from even a small number of PCs can impact MFSD2A expression in ECs. Using a PC-conditioned medium, we examined whether soluble factors spontaneously released from intact PCs were responsible for increased MFSD2A expression in ECs. However, conditioned medium derived from intact PCs alone failed to increase MFSD2A expression in ECs (Figure 1). These findings indicate that the PC-mediated enhancement of MFSD2A expression in ECs requires an initiating signal from ECs to PCs. Overall, these data suggest that PCs upregulate MFSD2A protein expression in ECs in response to EC-derived factors.

Next, we demonstrated that AG1296, an inhibitor of PDGFR, suppressed the PC-mediated increase in MFSD2A (Figure 1) and that *Pdgfrb*-knockdown PCs were unable to upregulate protein expression in ECs to the same extent as normal PCs (Figure 2). The results obtained using the PDGFRβ inhibitor and *Pdgfrb*-knockdown PCs suggest that brain endothelial PDGF-BB plays a vital role in the regulation of MFSD2A expression in ECs by PCs through EC–PC interactions mediated by PDGF-BB/PDGFRβ signaling. Furthermore, PCs increased the plasma membrane localization of the MFSD2A protein in ECs in correlation with its protein expression levels (Figure 3). This increase was inhibited by AG1296 (Figure 4). Collectively, these results suggest that PDGF-BB/PDGFRβ signaling between ECs and PCs mediates both MFSD2A protein expression and its plasma membrane localization in ECs.

Finally, we confirmed that the upregulation of MFSD2A protein expression and plasma membrane localization in ECs leads to functional improvement, indicated by the cellular uptake of DHA. Indeed, PC-induced DHA uptake by ECs was dependent on the number of PCs (Figure 5). As the specific localization of MFSD2A in BMECs plays a functional role in transport activity [6,7], an increase in DHA uptake probably occurred as a result of the PC-induced plasma membrane localization of MFSD2A.

PCs and BMECs can communicate through both direct contact and indirectly through ion exchange via gap junctions and other paracrine molecules to form and maintain the BBB structure [17,18,19,20]. However, the results of this study are not dependent on direct contact, as we used non-contact coculture models. Various ligand-to-receptor signals are transmitted between PCs and BMECs in the CNS microvasculature, involving the PDGF-BB/PDGFRβ, VEGF/VEGFR2, TGF-β/TGFBR, and angiopoietin 1/Tie2 signaling pathways, which contribute to angiogenesis, stabilization, and the remodeling of the mature vasculature [17,21,22]. PDGF-BB derived from ECs is reportedly crucial for appropriate PC recruitment to blood vessels [23,24,25]. Furthermore, we provide new evidence that PDGF-BB/PDGFRβ signaling between PCs and BMECs is involved in MFSD2A protein expression and membrane localization in BMECs.

Previous studies have revealed the intricate mechanisms regulating MFSD2A expression in the BBB [26,27]. Wnt signaling has been identified as a key player in the transcriptional regulation of MFSD2A expression [26], while the PTEN/AKT/NEDD4-2/MFSD2A axis controls BBB permeability [27]. These reports, along with our findings, suggest that PCs release mediators that activate Wnt signaling and the PTEN/AKT/NEDD4-2/MFSD2A axis in BMECs. This activation occurs through the interaction of PCs with BMECs via PDGF-BB/PDGFRβ signaling. However, further studies are warranted to fully elucidate the mechanism regulating MFSD2A expression and membrane localization in BMECs by PC-derived soluble factors.

PCs disappear from the cerebral capillary wall upon aging [28,29]. Furthermore, we have shown in previous studies that DHA transport across the BBB is reduced in correlation with the decreased MFSD2A expression levels in brain microvessels with aging [8]. Based on the present results, we hypothesize that age-related PC loss and degeneration may contribute to the decreased brain transport of DHA due to reduced MFSD2A expression. A previous study observed that DHA levels decreased in the BMECs of aged mice, corresponding to the age-associated downregulation of MFSD2A [30]. Moreover, MFSD2A dysfunction has been found to impair lipid homeostasis in BMECs [31]. Therefore, the age-related loss of PCs from brain microvessels may reduce BBB function and alter lipid homeostasis via reduced DHA levels in BMECs, leading to age-related BBB barrier dysfunction.

## 4. Materials and Methods

### 4.1. Animals

All animal experiments were approved by the Laboratory Animal Care and Use Committee of Fukuoka University [Permit No: 2204002 (Approved: 11 April 2022) and 2315117 (Approved: 19 March 2024)]. Male and female Wistar rats (3–4 weeks old) were purchased from Japan SLC, Inc. (Shizuoka, Japan) and housed under a controlled temperature (22 ± 2 °C) and 12/12 h light/dark cycle (lights on from 7:00 to 19:00), with ad libitum access to water and a chow diet. Animals were anesthetized with isoflurane and euthanized by decapitation. The brains were then removed from the skull and used for the primary cultures of rat brain ECs and PCs.

### 4.2. Primary Cultures of Rat Brain ECs and PCs

The primary cultures of rat brain ECs and PCs were established, as previously described [32,33]. Isolated brain microvessels were plated on 100 mm dishes coated with collagen type IV (0.1 mg/mL, Sigma, St. Louis, MO, USA; C5533) and fibronectin (0.075 mg/mL, Sigma; F1141-–5MG) to obtain ECs, which were maintained in EC medium (Dulbecco’s Modified Eagle Medium/F12 [DMEM/F12; Wako, Wakayama City, Japan, 042–30555] supplemented with 10% fetal bovine serum [FBS; Biosera, Kansas, MO, USA; FB-1365/500], basic fibroblast growth factor [1.5 ng/mL, R&D, Minneapolis, MN, USA; 2099-FB-025], heparin [100 μg/mL, Sigma; H3149], insulin [5 μg/mL], transferrin [5 μg/mL], sodium selenite [5 ng/mL; insulin–transferrin–sodium selenite media supplement, Sigma; I1884], penicillin [100 units/mL], streptomycin [100 µg/mL; penicillin–streptomycin mixed solution, Nacalai Tesque, Kyoto, Japan; 09367–34], and gentamicin [50 μg/mL, Biowest, Riverside, MO, USA; L0012]) containing puromycin (4 μg/mL, Nacalai Tesque; 14861–84) for 3 d at 37 °C in a humidified atmosphere with 5% CO_2_/95% air until they reached 70–80% confluency.

Primary cultured PCs were obtained by plating brain microvessels in uncoated 75-cm^2^ flasks and culturing in PC medium (DMEM [Wako, 048–29763] supplemented with 20% FBS [Gibco/Thermo Fisher Scientific, Waltham, MA, USA; A3160602], penicillin [100 units/mL], streptomycin [100 µg/mL; penicillin–streptomycin mixed solution, Nacalai Tesque, 09367–34], and gentamicin [50 μg/mL, Biowest, Riverside, MO, USA; L0012]) at 37 °C in a humidified atmosphere with 5% CO_2_/95% air for 9 d, with fresh culture media replaced once on the third day of culture. The cells typically reached 70–80% confluency by the end of this period.

### 4.3. Non-Contact Cocultured BBB Models Comprising Rat Brain ECs and PCs

ECs were seeded in a six-well plate (30 × 10^4^ cells/well), 24-well transwell inserts (5 × 10^4^ cells/transwell, 0.4 µm pore size; Corning, Midland, MI, USA; 3470), and 24-well plates (10 × 10^4^ cells/well) coated with collagen type IV (0.1 mg/mL, Nitta Gelatin Inc.; 638–05921) and fibronectin (0.025 mg/mL, Sigma), and maintained in EC medium supplemented with 500 nM of hydrocortisone (Sigma; H0135). PCs were seeded in six-well transwell inserts (3–12 × 10^4^ cells/transwell), 35 mm dishes (10 × 10^4^ cells/dish), a 24-well plate (2.5 × 10^4^ cells/well), and 24-well transwell inserts (1–4 × 10^4^ cells/transwell, 0.4 µm pore size; Corning; 3470) coated with Cellmatrix Collagen Type I–C (0.1 mg/mL, Nitta Gelatin Inc., Osaka, Japan) and cultured in PC medium. The non-contact cocultured BBB models, comprising ECs in the plate with PCs in transwell inserts or ECs in transwell inserts with PCs in the plate, were maintained in EC medium supplemented with 500 nM of hydrocortisone for 1–3 d.

### 4.4. Treatment of Brain ECs with Conditioned Medium Derived from PCs

PCs were cultured on 35 mm dishes in EC medium supplemented with 500 nM of hydrocortisone for 3 d, and the medium was collected in tubes, which served as the PC-conditioned medium. As a control, to obtain a cell-free conditioned medium, EC medium supplemented with 500 nM of hydrocortisone was incubated in 35 mm dishes without PCs for the same duration and collected in tubes. Subsequently, ECs were cultured in PC-conditioned or cell-free conditioned medium for 3 d in vitro.

### 4.5. Treatment with AG1296 for Evaluating MFSD2A Protein Levels

PCs and ECs cultured in six-well transwell inserts in six-well plates were treated with vehicle (0.1% dimethyl sulfoxide [Wako, 045–24511]) or AG1296 (10 µM; SantaCruz, Dallas, TX, USA; sc-200631) for 3 d in EC medium supplemented with 500 nM of hydrocortisone. As controls, ECs cultured without PCs were treated with vehicle or AG1296 (10 µM) for 3 d.

### 4.6. Small Interfering RNA (siRNA) Transfection

PCs cultured in 35 mm dishes or in six-well transwell inserts were transfected with Rat Pdgfrb Silencer^®^ Select Pre-designed siRNA (siPdgfrb, 50 nM; Life Technologies/Thermo Fisher Scientific, 4390771) or Silencer Select Negative Control (50 nM; Life technologies/Thermo Fisher Scientific, 4390843) using the Lipofectamine^®^ RNAiMAX Transfection Reagent (4 μL; Invitrogen/Thermo Fisher Scientific, 13778075) and cultured in PC medium for 1–3 d. Furthermore, PCs were transfected with siPdgfrb or negative control siRNA, as mentioned above, and cultured in PC medium for 1 d. Then, the PC medium was replaced with EC medium, and the cells were cultured for an additional 3 d. PDGFRβ protein expression in PCs was assessed using Western blot.

### 4.7. Non-Contact Cocultured BBB Models Comprising Brain ECs and PCs Transfected with siPdgfrb

PCs cultured in six-well transwell inserts were transfected with the lipid complex containing siPdgfrb or a negative control, cultured in PC medium for 1 d and then cocultured for 3 d with ECs grown in a six-well plate in EC medium supplemented with 500 nM of hydrocortisone. In parallel, as control groups, ECs were cultured without PCs for 3 d.

### 4.8. Treatment with AG1296 for Evaluating Plasma Membrane Localization of MFSD2A

PCs at the bottom of 24-well plates were pre-incubated with vehicle or AG1296 (10 µM) in EC medium supplemented with 500 nM of hydrocortisone for 15 min. These then were cocultured for 3 d with ECs grown in 24-well transwell inserts in the same medium containing vehicle or AG1296 (10 µM).

### 4.9. Western Blotting

ECs and PCs were scraped from the culture plates and lysed using a lysis buffer 10 mM Tris-HCl [pH 6.8; Nacalai Tesque; 35434–34], 100 mM NaCl [Sigma; 28–2270–5], 1 mM ethylenediaminetetraacetic acid [pH 8.0; Wako; 311–90075], 1 mM egtazic acid [Wako; 346–01312], 10% glycerol, 1% Triton-X100 [Sigma; X100], 0.1% sodium dodecyl sulfate [Nacalai Tesque; 02873–75], 0.5% sodium deoxycholate [Sigma; D6750], 20 mM sodium pyrophosphate [Sigma; S6422], 2 mM sodium orthovanadate [Sigma; S6508], 1 mM sodium fluoride [Wako; 196–01975], 1% protease inhibitor cocktail [Sigma; P2714], 1% phosphatase inhibitor cocktail 2 [Sigma; P5726], 1% phosphatase Inhibitor Cocktail 3 [Sigma; P0044], and 1 mM phenylmethylsulfonyl fluoride [Sigma; P7626]. The total protein concentration in cell lysates was determined using the Pierce™ BCA Protein Assay Kit (Thermo Fisher Scientific; 23225). Equivalent amounts of protein from each sample were electrophoretically separated on 7.5% TGX Stain-Free gradient acrylamide gels (Bio-Rad, Hercules, CA, USA; 161–0181) or 12% TGX Stain-Free acrylamide gels (Bio-Rad; 161–0185) and then transferred onto low-fluorescence polyvinylidene difluoride membranes (Bio-Rad; 1704274). Membranes were blocked using Blocking One (Nacalai Tesque; 03953–95) and then probed with antibodies against MFSD2A (1:1000; Sigma; SAB3500576), PDGFRβ (1:1000; Cell Signaling Technology, Danvers, MA, USA; 3169S), and β-actin (1:8000; Sigma; A1978). The membranes were then washed and incubated with horseradish peroxidase-conjugated goat anti-rabbit IgG (Bio-Rad; 170–6515) or goat anti-mouse IgG (Bio-Rad; 170–6516), as appropriate. Immunoreactive bands were detected using the Clarity Western ECL Substrate (Bio-Rad; 1705061). Images of the bands were digitally captured using MultiImager II ChemiBOX (BioTools, Gunma, Japan), and band intensities were quantified using ImageJ 1.54g software (National Institutes of Health Image, Bethesda, MD, USA). The relative intensity of each protein was expressed as the ratio of the protein normalized to β-actin.

### 4.10. Immunofluorescence Staining

ECs cultured on the membranes of 24-well transwell inserts were fixed in 99.8% methanol (Nacalai Tesque, 21915-93) for 2 min at −20 °C. ECs were permeabilized and blocked with 0.3% Triton™ X-100 (Sigma; X100) in Blocking One for 30 min and then incubated with primary antibodies against occludin (1:100; Invitrogen/Thermo Fisher Scientific; 33–1500) and MFSD2A (1:100; Sigma; SAB3500576) overnight at 4 °C. They were then incubated with CF488-labeled goat anti-mouse IgG (1:1000; Nacalai Tesque) for occludin and Cy3-labeled donkey anti-rabbit IgG (1:1000; Jackson ImmunoResearch, West Grove, PA, USA) for MFSD2A for 1 h at 20–25 °C. The membrane of the transwell inserts was cut and mounted on a slide glass with VECTASHIELD Mounting Medium containing 4′,6-diamidino-2-phenylindole (Vector Laboratories, Newark, CA, USA; H-1200). All samples were imaged using a fluorescence microscope (BZ-X710, KEYENCE, Osaka, Japan).

### 4.11. Cellular Uptake of [^14^C]DHA by Brain ECs Cocultured with PCs

Cellular DHA uptake was measured, as previously described [8]. ECs seeded in a 24-well plate were cocultured with or without PCs seeded in 24-well transwell inserts for 3 d. Subsequently, ECs were incubated with 0.2 mL of physiological buffer containing 0.1 µCi/mL [^14^C]DHA (incubation buffer; American Radiolabeled Chemicals, St. Louis, MO, USA; ARC0380) at 37 °C for 2 min. Subsequently, they were washed three times with D-PBS and incubated with 0.2 mL of 1 M NaOH (Wako; 192-02175) at 37 °C for 3 h. The total protein concentration in cell lysates was determined using the Pierce™ BCA Protein Assay Kit. Samples were added to 10 mL of a liquid scintillation cocktail (Pico-Fluor Plus; PerkinElmer; 6013699), and [^14^C]DHA radioactivity in the cell lysate was measured using a liquid scintillation counter. [^14^C]DHA uptake by ECs was expressed as a cell/medium ratio calculated by dividing the radioactivity per milligram of protein by the radioactivity per microliter of the incubation buffer.

### 4.12. Statistical Analysis

Results are expressed as the mean ± standard error of the mean. Statistical analyses were performed using GraphPad Prism 8.0 (GraphPad, San Diego, CA, USA). Unpaired *t*-tests were used to compare two groups. Statistical differences among groups were analyzed using one-way analysis of variance (ANOVA) followed by Dunnett’s multiple comparisons test. Statistical analyses were performed for two factors between groups using two-way ANOVA, followed by Šídák’s multiple comparisons test. Differences were considered statistically significant at *p* < 0.05. 

## 5. Conclusions

Our results demonstrate that PCs play a crucial role in enhancing both the expression and plasma membrane localization of MFSD2A in BMECs. This enhancement is achieved through PDGF-BB/PDGFRβ signaling between PCs and BMECs. Furthermore, the increased plasma membrane localization of MFSD2A may functionally activate MFSD2A as a DHA transporter. These findings open new avenues for understanding the relationship between PCs and MFSD2A under both physiological and pathological conditions; however, their potential implications remain to be fully explored and require further research.

## Figures and Tables

**Figure 1 ijms-26-05949-f001:**
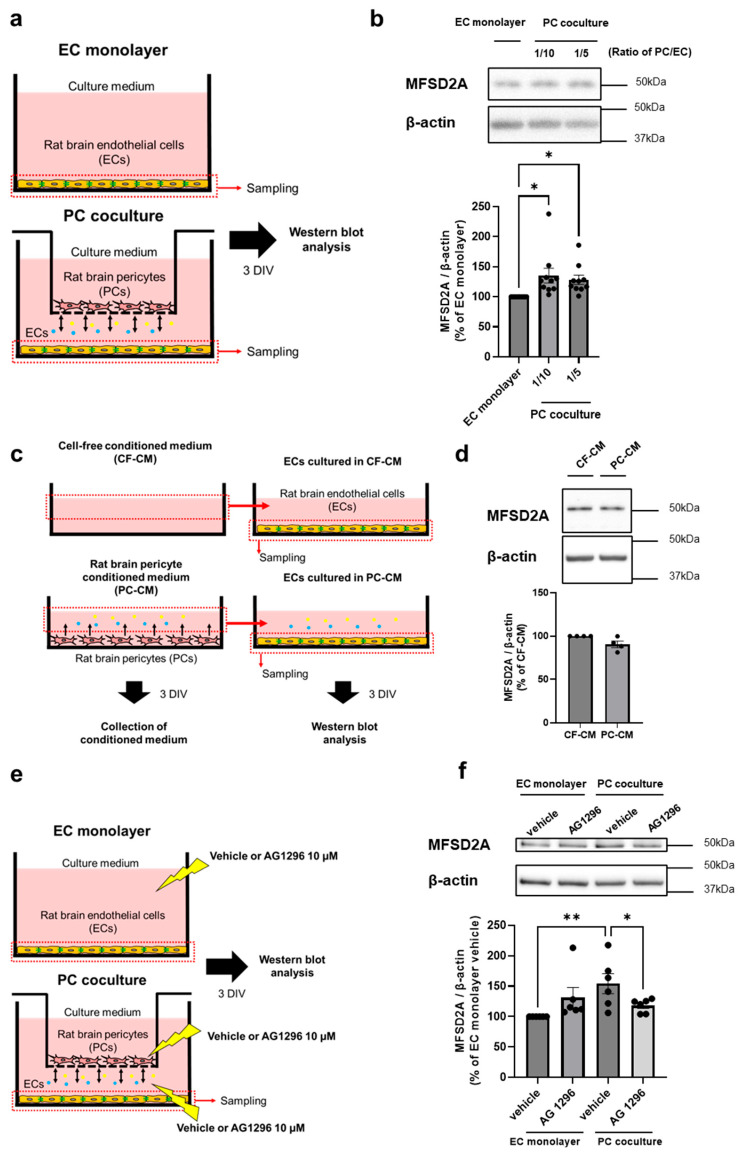
Impact of pericytes on MFSD2A expression in brain endothelial cells via cell-to-cell interaction. (**a**) Schematic representation of the experimental procedure using a non-contact cocultured BBB model comprising rat brain endothelial cells (ECs) and pericytes (PCs). ECs with or without PCs were cultured for 3 d in vitro (DIV). (**b**) Top: Representative Western blot images of MFSD2A and β-actin in ECs (30 × 10^4^ cells/well) cultured with or without PCs (3 and 6 × 10^4^ cells/well) for 3 DIV. Bottom: Quantification of band intensities normalized to β-actin as the loading control (*n* = 10). (**c**) Schematic diagram illustrating the collection of PC-derived conditioned medium and the subsequent treatment of ECs. (**d**) Top: Representative Western blot images of MFSD2A and β-actin in ECs cultured in pericyte-derived conditioned medium (PC-CM) or cell-free conditioned medium (CF-CM) for 3 DIV. Bottom: Quantification of band intensities normalized to β-actin as the loading control (*n* = 4). (**e**) Schematic diagram of non-contact cocultured BBB models treated with AG1296. ECs were cultured with or without PCs and simultaneously treated with vehicle or AG1296 10 µM, added to both chambers of the 6-well transwell plate for 3 DIV. (**f**) Top: Representative Western blot images of MFSD2A and β-actin in ECs from the EC monolayer and PC coculture, treated with or without AG1296 for 3 DIV (*n* = 6). Bars indicate the mean ± standard error of the mean. * *p* < 0.05 and ** *p* < 0.01, indicating significant differences compared to the corresponding EC monolayer or vehicle-treated group.

**Figure 2 ijms-26-05949-f002:**
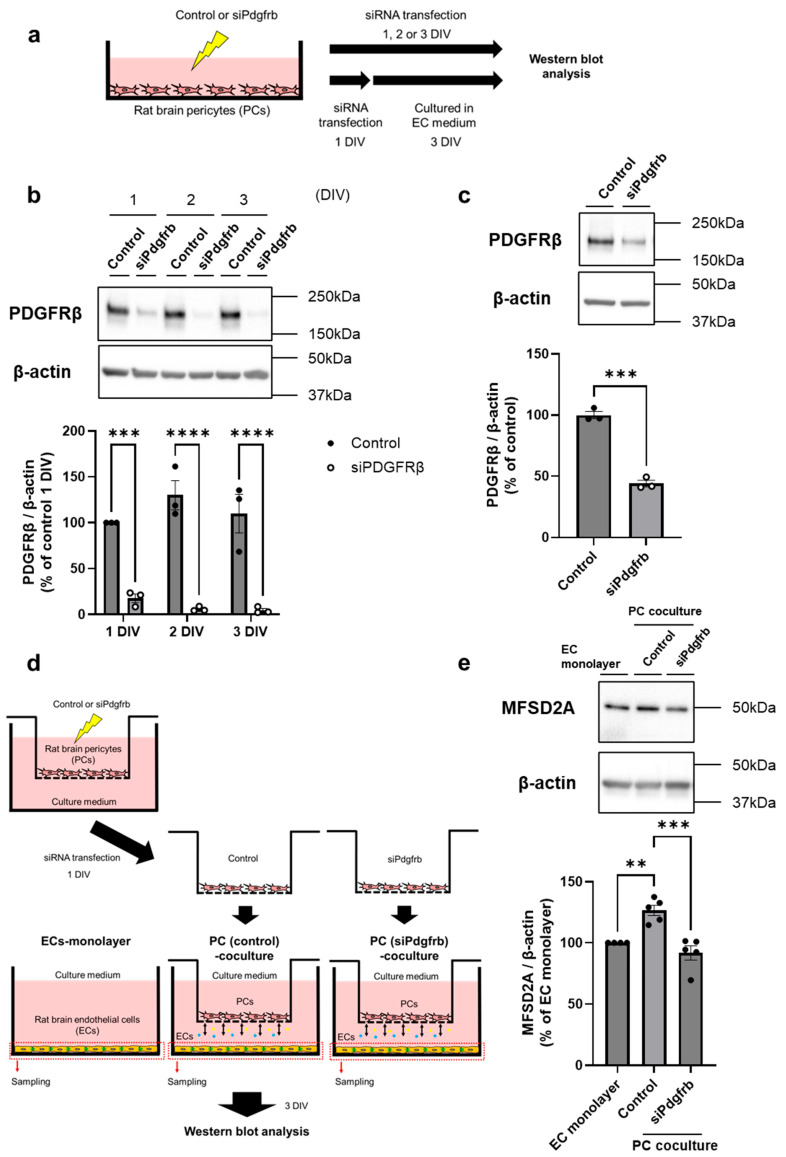
Effect of PDGFRβ-knockdown pericytes on MFSD2A expression in brain endothelial cells. (**a**) Schematic representation of the experimental procedure for siRNA transfection in rat brain pericytes (PCs). (**b**) Top: Representative Western blot images of PDGFRβ and β-actin in PCs transfected with siPdgfrb or negative control siRNA (Control) for 1–3 d in vitro (DIV). Bottom: Quantification of band intensities normalized to β-actin as the loading control (*n* = 3). (**c**) Top: Representative Western blot images of PDGFRβ and β-actin in PCs transfected with siPdgfrb or negative control siRNA (Control) for 1 DIV. Cells were treated with EC medium for an additional 3 DIV. Bottom: Quantification of band intensities relative to β-actin as the loading control (*n* = 3). (**d**) Schematic diagram illustrating the experimental procedure used to evaluate the effect of *Pdgfrb*-knockdown PCs on MFSD2A protein expression in ECs. (**e**) Top: Representative Western blot images of MFSD2A and β-actin in ECs cultured with or without PCs transfected with siPdgfrb or negative control siRNA (Control). Bottom: Quantification of band intensities relative to β-actin as the loading control (*n* = 4–5). Data are expressed as percentages relative to the control in the PC or EC monolayers. Bars indicate the mean ± standard error of the mean. ** *p* < 0.01, *** *p* < 0.001, and **** *p* < 0.0001; significant differences compared to the control or EC monolayer group.

**Figure 3 ijms-26-05949-f003:**
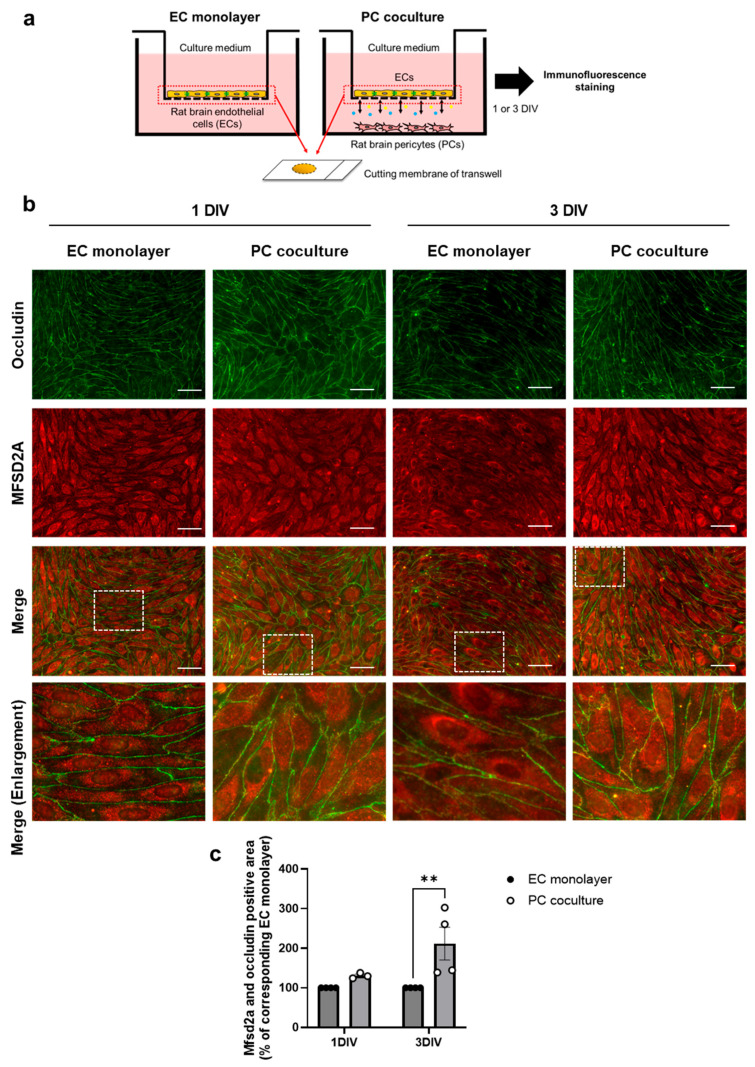
Effect of pericytes on MFSD2A plasma membrane localization in brain endothelial cells. (**a**) Schematic representation of the experimental procedure. ECs were cultured with or without PCs for 1–3 d in vitro (DIV) and subjected to immunofluorescence staining. (**b**) Representative fluorescence images showing staining of occludin (green), MFSD2A (red), and the merged images. The fourth row shows magnified views of dashed square regions in the merged images. (**c**) MFSD2A and occludin double-positive areas were detected and quantified. Data are expressed as the percentages relative to the corresponding EC monolayers (*n* = 3–4). Bars indicate the mean ± standard error of the mean. ** *p* < 0.01 indicates significant differences compared to each corresponding EC monolayer. Scale bar: 50 µm.

**Figure 4 ijms-26-05949-f004:**
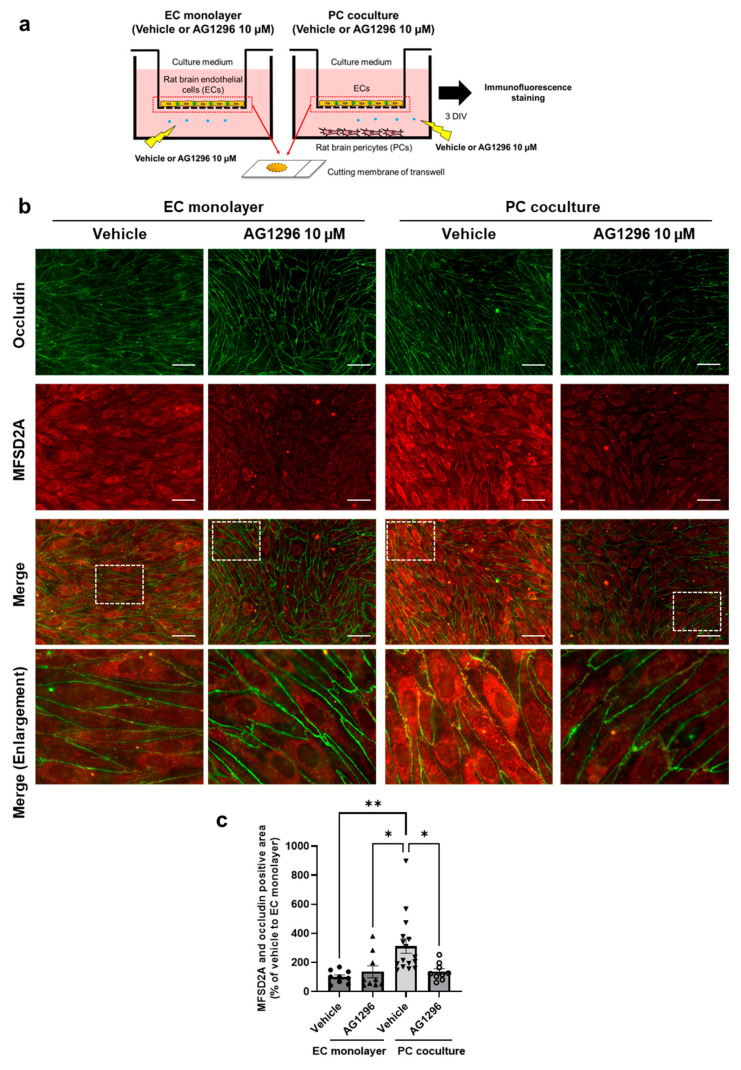
Impact of pericytes treated with AG1296 on MFSD2A plasma membrane localization in brain endothelial cells. (**a**) Schematic representation of the experimental procedure. Rat brain endothelial cells (ECs) were cultured with or without pericytes (PCs) and treated with the vehicle or 10 µM AG1296, added to the outer chamber of the 24-well transwell, for 3 d in vitro (DIV). Subsequently, ECs were subjected to immunofluorescence staining. (**b**) Representative fluorescence images showing immunostaining of occludin (green) and MFSD2A (red), along with merged images. The fourth row presents magnified views of dashed square regions in the merged images. (**c**) MFSD2A and occludin double-positive areas were the detected and quantified. Data are expressed as percentages relative to the vehicle-treated EC monolayer (*n* = 9–16). Bars indicate the mean ± standard error of the mean. * *p* < 0.05 and ** *p* < 0.01 indicate significant differences compared to the vehicle-treated PC coculture. Scale bar: 50 µm.

**Figure 5 ijms-26-05949-f005:**
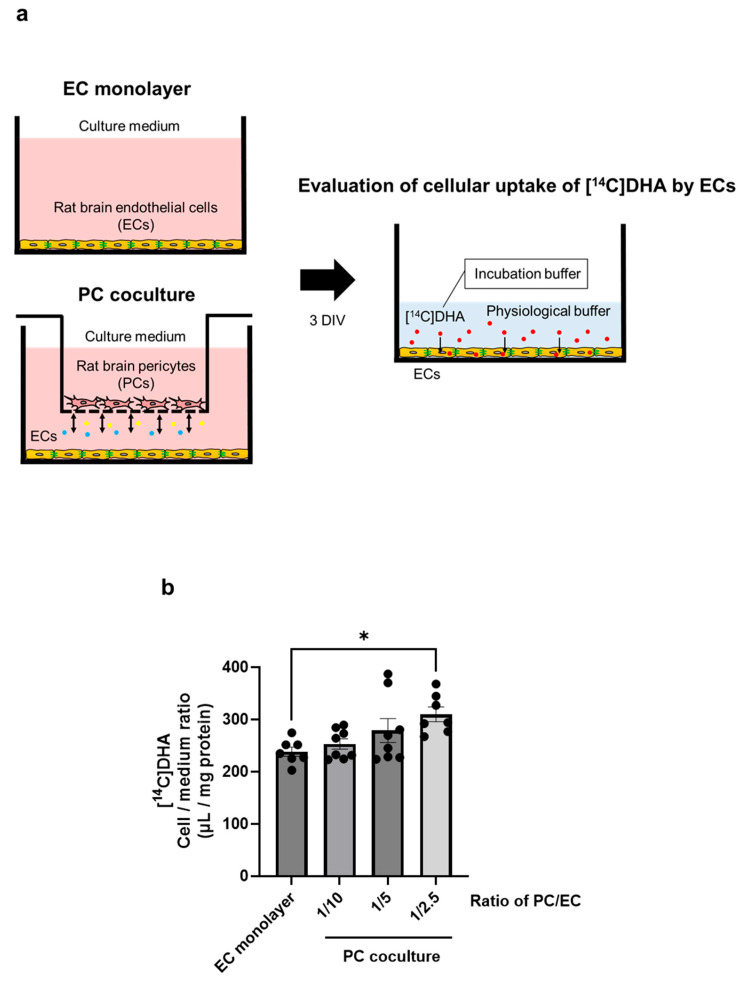
Effect of pericytes on the uptake of [^14^C]DHA by brain endothelial cells. (**a**) Schematic representation of the experimental procedure used to evaluate the uptake of [^14^C]DHA as non-esterified DHA by EC. ECs (10 × 10^4^ cells/well) were cultured with or without PCs (1, 2, and 4 × 10^4^ cells/well) for 3 d in vitro (DIV), followed by a cellular uptake assay. (**b**) The cellular uptake of [^14^C]DHA by ECs for 2 min is expressed as a cell/medium ratio (µL/mg protein). Data are shown as the mean ± standard error of the mean (*n* = 7–8). * *p* < 0.05 indicates significant differences compared to the EC monolayer group.

## Data Availability

The original contributions presented in this study are included within the article. Further inquiries can be directed to the corresponding author.

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
