# Peer review of "Brain Pericytes Enhance MFSD2A Expression and Plasma Membrane Localization in Brain Endothelial Cells Through the PDGF-BB/PDGFRβ Signaling Pathway"

_ijms, 2025, doi:10.3390/ijms26135949_

Round 1
Reviewer 1 Report
Comments and Suggestions for Authors
- The connection between PC and MFSD2A is unclear. Additional explanation would benefit new readers.
- The scale bar information is missing in the fluorescence images in Figure.
- The characterization of PC and EC is not provided. Please include the markers used for purity validation.
- A treatment period of 3 days seems too long. What is the rationale behind this selection? Was any shorter treatment duration considered?
- How was the knockdown validated? Were any experiments conducted to confirm its effectiveness?
Author Response
Reviewer 1
We appreciate the reviewer’s insightful comments and suggestions, which have helped us to significantly improve the quality of our manuscript. Given below is a point-by-response to all of the comments raised, and the changes have been incorporated into revised version of the manuscript.
Comment 1: The connection between PC and MFSD2A is unclear. Additional explanation would benefit new readers.
Response 1
We appreciate the reviewer’s suggestion. We have added more detailed sentences regarding the connection between pericytes and MFSD2A.
Page 2, Lines 56-59
Increased transcytosis was observed in cerebral vascular endothelial cells isolated from Mfsd2a knockout mice, in which pericytes appeared normal [6]. In contrast, pericyte-deficient mice exhibited a significant decrease in MFSD2A expression, indicating that pericytes are necessary for MFSD2A expression [6].
Comment 2: The scale bar information is missing in the fluorescence images in Figure.
Response 2
We appreciate the reviewer for pointing this out. We have added the scale bar information to the Figure legend.
Page 8, Line 185 and Page 10, Line 210
Scale bar: 50 µm.
Comment 3: The characterization of PC and EC is not provided. Please include the markers used for purity validation.
Response 3
We appreciate the reviewer’s suggestion. As shown below in Figure A, we validated the expression level of claudin-5, NG2, and GFAP as marker proteins for brain microvascular endothelial cells (ECs), brain pericytes (PCs), and astrocytes, respectively, using western blot analysis. These marker proteins were specifically expressed in ECs, PCs, and astrocytes isolated from Wistar rats. Additionally, we have already confirmed the expression of other characteristic markers at the protein, mRNA, and immunoreactivity levels in both the present and previous studies.
Figure A: Representative western blot images of NG2, GFAP, and claudin-5 in brain endothelial cells, pericytes, and astrocytes, prepared from Wistar rats.
Comment 4: A treatment period of 3 days seems too long. What is the rationale behind this selection? Was any shorter treatment duration considered?
Response 4
We appreciate the reviewer’s suggestion. A similar schedule has been employed to prepare in vitro BBB models. We have previously confirmed that brain endothelial cells (ECs) co-cultured with pericytes (PCs) exhibit enhanced barrier function, as evaluated by transendothelial resistance (TEER) and sodium fluorescein permeability, with peak performance observed after 3–5 days of co-culture. In addition, coculture of PCs and ECs was initiated when ECs were seeded into six-well plates. The protein yield from ECs was low after 1 day of coculture. Therefore, the coculture was performed for 3 days to ensure sufficient protein for analysis.
Comment 5: How was the knockdown validated? Were any experiments conducted to confirm its effectiveness?
Response 5
We appreciate the reviewer’s suggestion. In addition to Figure 2B and 2C which show that siPdgfrb transfection effectively reduced PDGFRβ in pericytes, we examined the impact of PDGFRβ knockdown on CD13 expression by siPdgfrb transfection, another pericyte marker, and found no changes in CD13 expression levels in PDGFRβ-knockdown pericytes. Furthermore, pericytes are known to enhance the expression of claudin-5, a tight junction-associated protein expressed in brain endothelial cells. To assess whether PDGFRβ signaling in pericytes contributes to this effect, we investigated the effect of PDGFRβ knockdown in pericytes on claudin-5 expression in brain endothelial cells. We confirmed that the pericyte-induced increase in claudin-5 expression was suppressed by PDGFRβ knockdown (Figure B).
Figure B: (a) The top panel shows representative western blot images of CD13 and β-actin in PCs transfected with siPdgfrb or negative control siRNA (Control) for 1 d in vitro (DIV). The bottom panel presents quantified band intensities normalized to β-actin as the loading control (n = 3). (b) The top panel shows representative western blot images of claudin-5 and β-actin in ECs cultured with or without PCs transfected with siPdgfrb or negative control siRNA (Control). The bottom panel presents quantified band intensities relative to β-actin as the loading control (n = 4–5). Data are expressed as percentages of the control in the PCs or EC monolayer. Bars indicate the mean ± standard error of the mean. **P < 0.01, significant differences compared to the control or EC monolayer group.

Reviewer 2 Report
Comments and Suggestions for Authors
In this manuscript the authors presented experimental evidence for the interaction between pericytes and endothelial cells from rat blood-brain barrier. The investigation provided some interesting insights into the regulation of the transporter MFSD2a in the endothelial cells. However, there are a few questions to be clarified:
- The authors didn't provide any rationale how they came to the idea that the PDGF pathway might regulate the expression and localization of MFSD2a
- The selectivity of the PDGFR inhibitor A1296 was not sufficiently addressed. According to the web site of the vendor, it does inhibit some other receptors
- Is it possible to stimulate the pericytes with PDGF ligands or activators, and then collect the conditioned medium for the stimulation of the endothelial cells?
- In figure 5B, uptake of DHA could be only shown with a PC coculture at a 1:2.5 ratio. However, this coculture ratio was not characterized in Figure 1.
Author Response
Reviewer 2
We appreciate the reviewer’s insightful comments and suggestions, which have helped us to significantly improve the quality of our manuscript. Given below is a point-by-response to all of the comments raised, and the changes have been incorporated into revised version of the manuscript.
Comment 1: The authors didn't provide any rationale how they came to the idea that the PDGF pathway might regulate the expression and localization of MFSD2a
Response 1
We appreciate the reviewer’s suggestion. We have added the following sentences to clarify the rationale that the PDGF pathway may regulate the expression and localization of MFSD2A.
Page 2, Lines 82-85
A previous study showed that MFSD2A expression dramatically decreased in Pdgfbret/ret mice, which exhibits a major loss of pericyte coverage [6]. Therefore, we hypothesized that PDGFRβ, as a receptor for PDGF-BB released from ECs, plays a critical role in regulating MFSD2A expression.
Page 6, Lines 162-163
A previous study using immuno-electron microscopy confirmed that MFSD2A is localized to the plasma membrane of cerebral cortex capillaries [6].
Comment 2: The selectivity of the PDGFR inhibitor A1296 was not sufficiently addressed. According to the web site of the vendor, it does inhibit some other receptors
Response 2
We appreciate the reviewer for pointing this out. We have added the following sentences to clarify the limitations of what can be investigated using AG1296.
Page 4, Lines 117-120
Although AG1296 is a potent and selective inhibitor of PDGFR kinase, it may also inhibit other kinases, including PDGFRα, Bek (FGF receptor) tyrosine kinase, and c-kit, in addition to PDGFRβ. Therefore, we further investigated whether PDGFRβ in pericytes contributes to regulating MFSD2A expression in ECs.
Comment 3: Is it possible to stimulate the pericytes with PDGF ligands or activators, and then collect the conditioned medium for the stimulation of the endothelial cells?
Response 3
We appreciate the reviewer’s suggestion. We speculate that conditioned medium collected from pericytes stimulated via PDGFRβ activation by PDGF ligands or other activators can increase MFSD2A expression levels in brain endothelial cells. Therefore, we agree that the experiment suggested by the reviewer is feasible, and needs to be further investigated.
Comment 4: In figure 5B, uptake of DHA could be only shown with a PC coculture at a 1:2.5 ratio. However, this coculture ratio was not characterized in Figure 1.
Response 4
We appreciate the reviewer for pointing this out. In Figure 1B, PC cocultures at ratios of 1:10 and 1:5 resulted in increased Mfsd2a expression in ECs. Therefore, the PC coculture at a 1:2.5 ratio is also expected to induce a similar increase in Mfsd2a expression. Indeed, as shown in Figure 5B, DHA uptake by ECs increased with the number of cocultured PCs. DHA uptake by ECs at the 1:5 ratio increased, although the difference was not statistically significant. In this study, the BBB models used for evaluating MFSD2A protein expression (Figure 1) and DHA uptake assay (Figure 5) differed in the seeding density of the cells due to the use of different well type of Transwell inserts (6-well and 24-well types, respectively). Although the conditions of BBB model using 6-well type inserts was not completely identical to those of the 24-well type BBB models, the enhanced DHA uptake by ECs observed in Figure 5B is likely attributable to increased Mfsd2a expression and its membrane localization by PCs.
Round 2
Reviewer 1 Report
Comments and Suggestions for Authors
Author has addressed all the comments. Thank you.
Author Response
Comment 1: Author has addressed all the comments. Thank you.
Response 1:
We appreciate the reviewer’s insightful comments and suggestions, which have helped us to significantly improve the quality of our manuscript.
Reviewer 2 Report
Comments and Suggestions for Authors
In section 2.5, the authors should add a note that, despite a significant increase of MFSD2A expression at a coculture ratio of 1:5, no significant increase of uptake of DHA could be observed. The increased uptake was only detectable at a coculture ratio of 1:2.5.
Author Response
Reviewer 2
We appreciate the reviewer’s suggestion, which has enabled us to considerably improve our manuscript. As indicated in the following response, we have taken the reviewer’s suggestion into account and revised the manuscript accordingly.
Comment 1: In section 2.5, the authors should add a note that, despite a significant increase of MFSD2A expression at a coculture ratio of 1:5, no significant increase of uptake of DHA could be observed. The increased uptake was only detectable at a coculture ratio of 1:2.5.
Response 1:
We appreciate the reviewer’s suggestion. In accordance with the reviewer’s suggestion, we have added the following sentence to Section 2.5.
Page 10, Lines 219-222
Despite a significant increase in MFSD2A expression at a PC:EC ratio of 1:5 as shown in Figure 1b, no significant increase in [14C]DHA uptake by ECs was observed. The significantly increased uptake was only detectable at a PC:EC ratio of 1: 2.5.